# Rapid and robust assembly and decoding of molecular tags with DNA-based nanopore signatures

Kathryn Doroschak [1], Karen Zhang [1], Melissa Queen [1], Aishwarya Mandyam[1], Karin Strauss [2], Luis Ceze [1] & Jeff Nivala [1]✉

Molecular tagging is an approach to labeling physical objects using DNA or other molecules that can be used when methods such as RFID tags and QR codes are unsuitable. No molecular tagging method exists that is inexpensive, fast and reliable to decode, and usable in minimal resource environments to create or read tags. To address this, we present Porcupine, an end-user molecular tagging system featuring DNA-based tags readable within seconds using a portable nanopore device. Porcupine's digital bits are represented by the presence or absence of distinct DNA strands, called molecular bits (molbits). We classify molbits directly from raw nanopore signal, avoiding basecalling. To extend shelf life, decrease readout time, and make tags robust to environmental contamination, molbits are prepared for readout during tag assembly and can be stabilized by dehydration. The result is an extensible, real-time, high accuracy tagging system that includes an approach to developing highly separable barcodes.

[1] Paul G. Allen School of Computer Science & Engineering, University of Washington, Seattle, WA 98195, USA. [2] Microsoft Research, Redmond, WA 98052, USA. ✉email: jmdn@cs.washington.edu

Tagging physical objects has proven useful for a range of formats and scenarios like UPC barcodes in packaging, QR codes for easy association of digital information with printed material, and radio-frequency identification (RFID) tags for inventory tracking. However, these tags cannot be applied to objects that are too small, flexible, or numerous, or in scenarios where the code should be invisible to the naked eye, like anti-forgery. Molecular tags address these shortcomings via their nanoscale footprint and difficulty to forge. However, existing methods for encoding digital information in molecules; including silica-encapsulated DNA tracers[1], DNA embedded in 3D printed material[2], microbial barcodes[3], or spatially isolated marker peptides[4]; require access to specialized labs and equipment to make new tags — making them impractical in applications that require a very large number of tags. In most cases, the protocol prevents real-time use cases, for example, in the case of PCR-based or SHERLOCK-based detection which takes tens of minutes in the best case[5]. An ideal molecular tagging system should be inexpensive and reliable, with fast readout and user-controlled encoding and decoding from end-to-end with minimal reliance on lab equipment.

Molecular biologists have also long been interested in tagging biomolecules, for example, with multiplexing samples in a single sequencing run through the inclusion of a sample-specific short DNA tag appended to each sequenced fragment. For nanopore sequencing in particular, direct raw signal techniques have improved efficiency of dereferencing multiplexing barcodes. DeepBinner classifies raw signals from 12 of Oxford Nanopore Technologies' 96 commercially available barcodes, improving upon their previous sequence-based tool, Porechop[6]. Similarly, DeePlexiCon classifies four hand-designed DNA barcodes for multiplexing in direct RNA sequencing, for which barcodes were not previously commercially available[7]. While these tools have enabled higher yields of demultiplexed reads, or in the case of RNA, made multiplexing possible, they would likely perform even better if the barcodes were designed to optimize raw signal separability. Other tools have also been developed for raw signal processing later in the sequencing pipeline, for example for genomic alignment[8].

Porcupine addresses limitations in prior DNA-based tagging methods by building on recent advances in nanopore-based DNA sequencing technologies and raw signal processing tools. The development of portable, real-time nanopore sequencing[9], together with new methods that simplify the modular assembly of predefined DNA sequences[10], creates additional opportunities for rapid writing and on-demand readout in low resource environments.

Porcupine is a molecular tagging system that uses synthetic DNA-based tags and nanopore-based readout. Porcupine encodes a 96-bit digital tag through the presence or absence of 96 pre-determined DNA fragments with highly separable nanopore signals, which we call molecular bits (or molbits, previously coined by Cafferty et al.[4]), shown in Fig. 1a. Although DNA is typically considered expensive for reading and writing, Porcupine lowers the cost by presynthesizing the DNA, which can then be mixed arbitrarily to create new molecular tags. Molecular tags are then read out quickly using a portable, low-cost sequencing device (Oxford Nanopore Technologies' MinION; Fig. 1b). Typically, raw nanopore signal must first be converted back to a DNA sequence in a computationally expensive process called basecalling, but we classify molecular tags directly from raw nanopore signal, forgoing basecalling. Raw signal classification is often used for DNA and RNA sample demultiplexing, which also uses DNA barcodes[6,7]; Porcupine drastically increases the number of barcodes by custom designing them to produce unique ionic current signatures. Error correction is also added to the tag

to resolve decoding errors, similar to electronic message transmissions systems and recently used in other molecular applications[11]. Molbits are prepared for readout (sequencing) prior to tag application and can be stabilized by dehydration, an approach that extends tag shelf life, decreases decoding time, and reduces contamination from environmental DNA. Thus, creating tags requires the set of pre-prepared molbits, sequencing adapter kit and its corresponding requirements (e.g., centrifuge, thermal cycler, and rotator mixer), and a dehydration method; and reading tags only requires the tag, nuclease-free water, and a MinION device. The result is a highly accurate real-time tagging system that includes an approach to developing highly separable barcodes. These barcodes, and the methods we use to develop them, are extensible; they can be used both within Porcupine to tag physical objects and beyond this system for other molecule-level tagging needs like sample multiplexing for nanopore sequencing.

## Results

**Molbit definition and assembly.** To develop Porcupine, we first defined an individual molbit as a DNA strand that combines a unique barcode sequence (40 nt) with a longer DNA fragment selected from a set of sequence lengths that we pre-determined (Fig. 2a). To make assembly of molbits simple and modular, we designed them to be compatible with Golden Gate Assembly, a convenient and scalable one-pot DNA assembly method that combines a TypeIIS restriction enzyme and ligase, by incorporating a short single-stranded overhang[10]. To increase classification accuracy and decrease computation time, we further optimized them to avoid basecalling. Thus, for the barcode region, the objective was to produce a large set of sequences that could generate unique ionic current signatures ("squiggles") to promote unambiguous classification.

**Designing highly separable molbits.** To model the predicted ionic current signature for arbitrary DNA sequences, we used "Scrappie squiggler" a tool that converts sequences of bases to ionic current via a convolutional model. For example, to demonstrate Scrappie's ability to accurately model real nanopore squiggles, we hand-designed a DNA sequence that appears as the letters "UW" in squiggle space (Fig. 2b), with high visual similarity to the simulated squiggle (except for noise). Scrappie's output also let us compute the signal similarity of two sequences quantitatively using dynamic time warping (DTW) as the distance measure. We used this approach inside an evolutionary model designed to make barcodes as separable as possible (Fig. 2c).

To produce a set of 96 orthogonal molbit barcode sequences, we initialized the evolutionary model using 96 random or pre-seeded starting sequences (see the Methods section). We perturbed each sequence independently in random order by mutating two adjacent nucleotides simultaneously at a random location. If the mutated sequence failed to improve the minimum and average DTW similarities between itself and all other sequences, we reversed the mutation and attempted again for the same sequence. We also restricted sequence similarity and free energy of the sequences to avoid labeling ambiguities and secondary structure (see the Methods section). Using this method, we began with a set of starting sequence that had a minimum DTW similarity of 2.9 and mean 4.2 ± 0.4 and achieved a final minimum of 4.2 and mean 5.8 ± 0.8 after 31 rounds of evolution (Fig. 2d), representing a ~40% improvement in both the minimum and mean.

**Using length as an additional encoding channel.** Given the set of 96 designed molbit barcodes, we wanted to increase the number of available molbits without requiring additional barcode

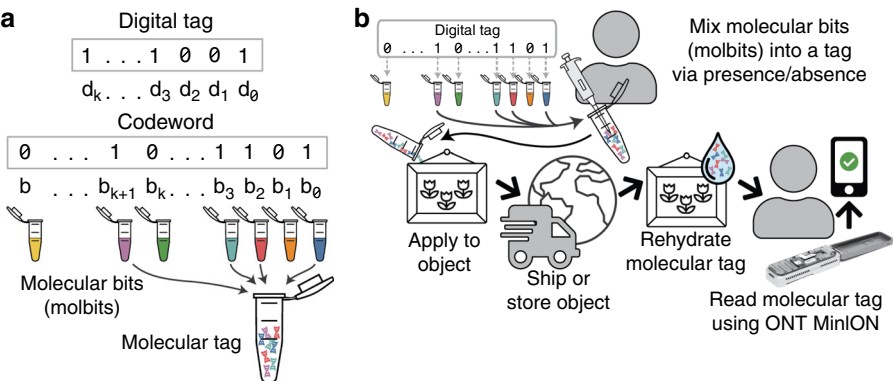

**Fig. 1 Creating a molecular tag using the Porcupine molecular tagging system. a** Porcupine's encoding scheme. A digital tag is converted into a codeword to add additional bits for error correction. Each codeword bit is assigned to a unique molbit, where 1s and 0s are represented by the presence or absence of individual molbits in the molecular tag mixture. **b** A user first defines a digital tag as a binary 96-bit number, and pipettes 1-bits into the molecular tag. The tag is applied to an object, which is then shipped or stored. To read the tag, it is rehydrated and loaded directly onto Oxford Nanopore Technologies' MinION device. Software then decodes the molecular tag without knowledge of the original digital tag.

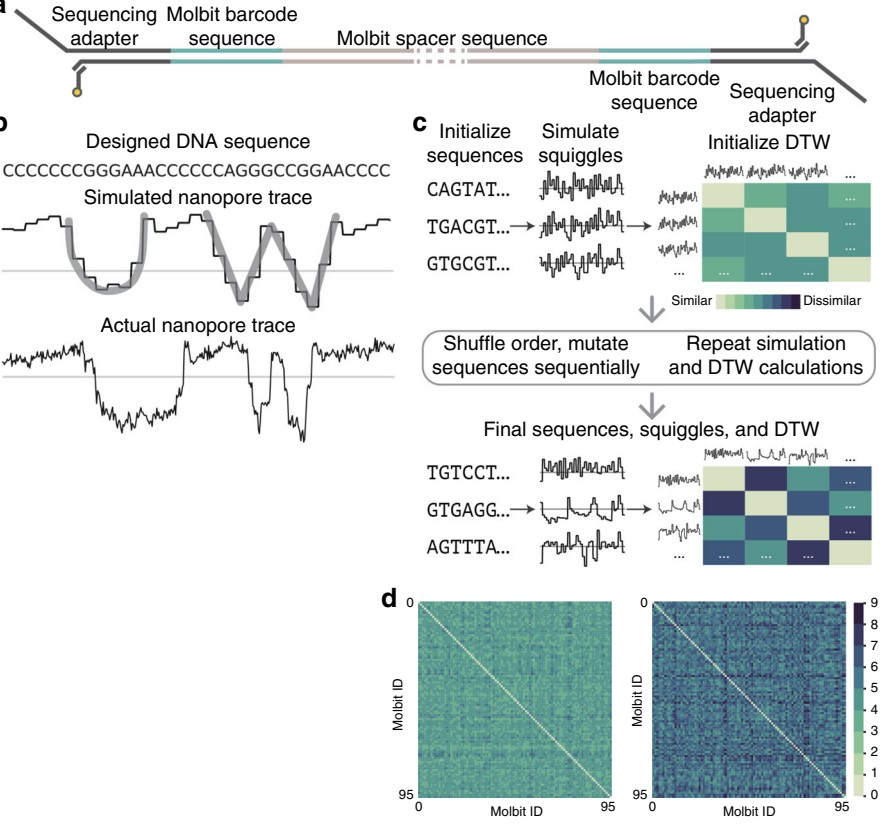

**Fig. 2 Molbit design scheme. a** Molecular bit (molbit) structure. The molbit sequence is attached to a spacer sequence via Golden Gate assembly to achieve a minimum length for sequencing and provide an additional encoding channel. Since the sequencing adapter is attached to both ends, the strand can be sensed from either direction. **b** The letters "UW" depicted visually in nanopore raw data (as opposed to encoded in the sequence contents). From top to bottom, the shown sequence was simulated using Scrappie and sequenced on the ONT MinION, demonstrating the viability of using simulations for designing intentional, arbitrary raw signal shapes. **c** Evolutionary model workflow. Each round of evolution begins with a set of sequences, their simulated squiggles, and pairwise Dynamic Time Warping (DTW) distances. The sequence order is randomized, and sequences are mutated one at a time, verifying DTW improvement (minimum and mean) after each attempt. **d** Dynamic time warping (DTW) scores before (left) and after (right) 31 iterations of the evolutionary model. After initialization, the minimum DTW similarity was 2.9 (mean 4.2 ± 0.4), and after evolution the minimum was 4.2 (mean 5.8 ± 0.8). Source data are provided as a Source Data file.

design or synthesis. To do this, we inserted a DNA fragment between the barcode regions as a spacer sequence, which can be set to different lengths as an additional encoding channel. Thus, since each molbit consists of the unique combination of a molbit

barcode plus a specific spacer sequence length, adding another length effectively adds an additional 96 molbits. Length works as an additional encoding channel because even without basecalling, the length of nanopore signals can be easily distinguished; the

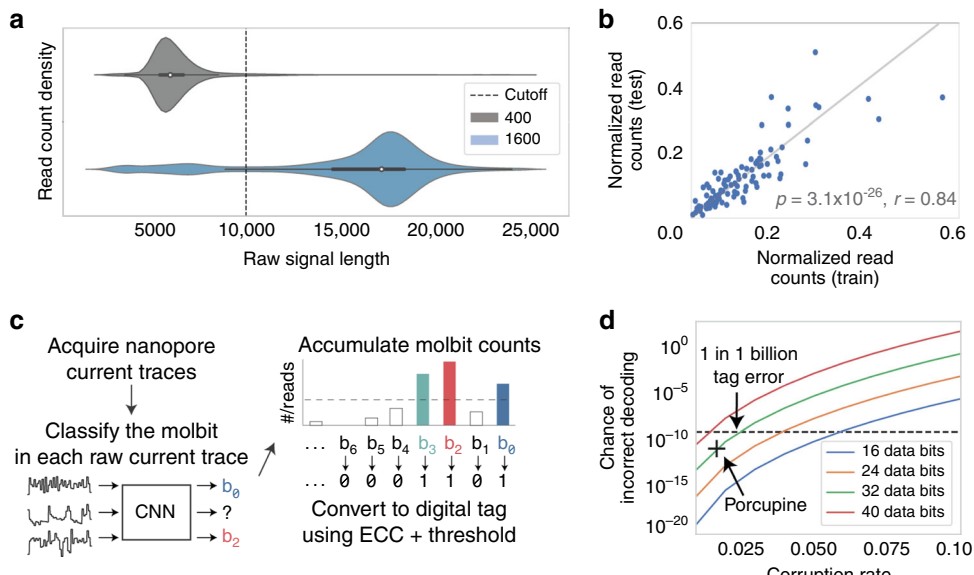

**Fig. 3 Molbit classification and tag decoding results. a** Violin plots with embedded box-and-whiskers showing the distribution of raw nanopore signal lengths for DNA sequences of length 400 nt ($n = 354,000$ reads) and 1600 nt ($n = 375,000$ reads). The white dot represents the median, and the thick center line shows the quartiles. The dashed vertical line shows the cutoff used when calling reads as 400 mers or 1600 mers. **b** Correlation of read counts for each molbit, demonstrating consistency in molbit occurrences between training and testing runs (two-sided Pearson correlation, $p = 3.1 \times 10^{-26}$, $r = 0.84$). Counts were first normalized within each run and normalized again after combining runs for either training or testing. **c** Tag decoding workflow, with error correcting codes (ECC). After acquiring nanopore current traces from a standard sequencing run, the molbit in each trace is identified using the CNN (confidence $\geq 0.9$). Successfully identified molbits are accumulated and converted into binary using a threshold for presence. This threshold is varied as error correction is carried out multiple times, accepting the binary digital tag that has the fewest differences from the received codeword. **d** Chance of incorrect tag decoding as a function of the bit corruption rate and number of data bits. This chance increases exponentially as the corruption rate and number of data bits increase linearly. The dashed line represents the goal of 1 in 1 billion tags incorrect, and the "+" marks Porcupine's chance of incorrect decoding. Source data are provided as a Source Data file.

signal length is roughly proportional to the DNA fragment length. We tested this concept using two spacer lengths (400 and 1600 nucleotides extracted from an arbitrary portion of plasmid pCDB180; see the Methods section) and found that simple signal length binning was sufficient for decoding. The median signal length for the 400 and 1600 nt strands was 5768 and 16,968, respectively, and a cutoff at 9800 gave 91% accuracy, where most errors were caused by long strands misidentified as short strands (Fig. 3a). We additionally evaluated the source of the misclassifications, particularly with the 1600 nt strands. (Supplementary Methods) For remaining experiments, we focused on a single spacer length to reduce experimental complexity, and develop the more conceptually interesting molbit design.

**Classifying molbits directly from raw nanopore signals.** After designing the molbits and acquiring a theoretical understanding of their separability, we developed a convolutional neural network (CNN) model to accurately classify them directly from real raw nanopore data (Supplementary Fig. 1). Direct signal classification is preferred over sequence-based methods primarily because classification is an inherently simpler problem than full sequence decoding, where the problem can be reduced to distinguishing between 96 distinct signals rather than reproducing the exact series of underlying nucleotides generating an arbitrary signal. This enables simpler model architectures, which typically come with lower computational and training data requirements. In our model, molbits do not need to be segmented or otherwise isolated from the raw data, rather, the CNN uses only the first portion of the each molbit since the molbit barcode is located at the beginning of the strand. We gathered training data by dividing the 96 molbit barcodes into six sequencing sets, with each molbit appearing once; optimizing the sets for maximum sequence

separability to improve labeling (see the Methods section); and running each set on the MinION. We assigned labels to each molbit read using traditional basecalling methods and a modified semilocal Smith–Waterman sequence alignment[12], using only high-confidence alignments (see the Methods section). For test data, we divided the 96 molbits into two sequencing sets, with each molbit appearing once, and gathered the data as described for training. The model was trained for 108 iterations, with a final training, validation, and test accuracy of 99.9%, 97.7% and 96.9%, respectively, compared to sequence-derived labels. However, in real world decoding, all reads are classified, not just those that pass basecalling and sequence alignment. We found that the CNN was consistently able to confidently classify a larger portion of the reads (97% of reads in the test set) than basecalling plus alignment (75% of reads in the test set), revealing a large portion of reads that we could not easily validate. We reasoned that if the occurrence of each molbit was proportional between basecalling and the CNN, that the CNN was likely not making spurious calls, but was perhaps performing better on the raw signal data. Read counts correlate extremely well between the two methods (Supplementary Fig. 2), revealing another advantage of direct signal classification, making use of ~30% more data than sequencing alone. As a result, "accuracy" only reflects the model accuracy and does not necessarily measure each molbit's error rate or the overall chance of decoding the tag incorrectly.

**Encoding data in molecular tags with error correction.** We next composed actual molecular tags. We assigned each molbit a unique position in a binary tag, allowing each 1 or 0 to represent the presence or absence of a specific molbit over the course of a single tag sequencing (decoding) run. To determine presence or absence, we used CNN-classified read counts for each molbit.

Ideally, 0 bits would have zero reads, and 1 bits would have nonzero reads. However: two factors complicated our setting this threshold to determine bit presence: (1) nonzero read counts for molbits not present in an experiment, and (2) significant variations in counts for molbits present in an experiment, which we found to be up to 20–30× in our training and test sets (Supplementary Methods). Fortunately, these variations were consistent when we compared the ratios of molbit counts in the training and test data (Fig. 3b). We accounted for this variation by scaling all read counts by a fixed vector based on these ratios. Thresholding and scaling read counts reduced our per-bit error rate from 2.9 ± 1.8% to 1.7 ± 1.6%, a 42% reduction.

Since a reliable tagging system should have a very low chance of incorrect decoding (e.g., 1 in 1 billion), we decided to further reduce our overall tag decoding error rate by including error correcting codes (ECCs) as part of our tag design (Fig. 3c). The simplest nonECC method for encoding information in these tags is a naive 1:1 mapping between digital bits and molbits; however, with this method, even a single bit error makes the tag unrecoverable (i.e., produces an incorrect decoding). In our system, bits are set to 1 or 0 using a threshold for presence or absence on the read counts, meaning that any 0-bits above this threshold are instead flipped to 1, and vice versa. ECCs reduce the possibility of unrecoverable tags despite the relatively high per-bit error rate by reserving a smaller number of bits for the digital message and creating a codeword by projecting this message into a larger space with greater separability. This allows more bits to be flipped before the message is decoded incorrectly. To encode the digital message, we simply multiply the message by a binary matrix of random numbers, known as a random generator matrix (Supplementary Methods). The number of bits reserved for the ECC depends on the application's error tolerance and the per-bit error rate (Fig. 3d). As the error rate increases, the chance of incorrect decoding increases exponentially. Thus, the number of bits for the message must be chosen carefully. We chose a message size of 32 bits, which at an error rate of 1.7% produces a $1.6 \times 10^{-11}$ chance of incorrect decoding and permits ~4.2 billion total unique tags, with correct decoding guaranteed at or below nine bit errors.

**End-to-end encoding and decoding**. Next, as a proof-of-principle for our tagging system, we demonstrated end-to-end tag encoding and decoding of the acronym "MISL" short for "Molecular Information Systems Lab" (Fig. 4a). We began by encoding MISL into binary using ASCII, which uses 8 bits for each character, for a total of 32 bits, and we multiplied this bit vector by the generator matrix to produce a 96-bit codeword. The molecular tag was then prepared as explained previously, with one modification for lab efficiency (see the Methods section). Once the molecular tag was assembled, it was prepared for sequencing and read out using an ONT MinION. We then identified the molbits from the raw data using our trained CNN classifier, accumulated a count for each molbit, and rescaled these counts (as explained above) to accommodate systematic read count variances. We then decoded the tag as described for the ECC by binarizing the counts using a sliding read count threshold to determine presence and absence, then finding the distance between the binarized counts and the nearest valid codeword at each read count threshold. When the edit distance is low enough to guarantee a unique correct decoding (this distance is 9 for this ECC), the molecular tag decoding is complete. The earliest correct decoding occurred less than 7 s after sample loading (109 molbit strands observed), demonstrating reliable encoding and decoding of a 32 bit message in only a few seconds using a portable sequencing instrument.

**Measuring decoding time**. Finally, to robustly estimate correct decoding time under these conditions, we simulated new tags using our original two test runs and one additional random tag by reassigning molbit labels within the tags (e.g., molbit 0 in the original tag is given a new label, molbit 42, in the simulated tag). This simulation impacts the error rate both positively and negatively due to the nonuniform distribution of distances between codewords in the ECC. To achieve this remapping, we generated new codewords with the same number of 1-bits as the original tag data sets. The original observed molbits were then randomly assigned to the new bit ordering: 1-molbits in the original tag were assigned new 1-molbit labels in the synthetic tag. After creating these new codewords, we sampled reads without replacement, accumulated a count for each molbit, and decoded as above, with ten repetitions per run time, where run time was estimated by the average of 10,000 reads per minute. We found that some tags could be successfully decoded with only a few seconds of sequencing data, and all tags after just 10–15 s (Fig. 4b).

## Discussion

In summary, Porcupine offers a method for molecular tagging based on the presence or absence of synthetic DNA sequences that generate explicitly unique nanopore raw current traces. By directly manipulating segments of nanopore raw current and keeping unique sequences short, we reduce synthesis costs for the end user and produce visually unique nanopore current traces, enabling high accuracy decoding. The speedy decoding time means our system can decode using newer technologies such as the Flongle, a cheaper, single-use flowcell produced by ONT with a quarter of the pores of the MinION in as little as 1–3 min. In addition, tags can be prepared for sequencing at the time of tag creation and then shelf-stabilized by dehydration, further reducing readout time at the cost of increasing "writing" time through sequencing preparation. Prepreparation appears to have minimal risk, supported by correct tag decoding and minimal changes to sequencing yield, $Q$-score, and basecalled sequence length when sequencing a freshly prepared and dehydrated tags (at 0 and 4 weeks from dehydration (Supplementary Fig. 3). In the future, more bits can be acquired by adding more insert lengths, by extending the length of the unique region to allow more variation between molbits, or by combining barcode regions serially. Furthermore, a generative model for molbit design may be a natural next step, especially if a dramatically larger number of molbits is desired, because the evolutionary model computations scale exponentially with the number of desired bits.

One limitation of Porcupine is the variation in molbit counts. Although decoding remains robust to bit errors, more stability in read counts would increase the amount of information we could encode by reducing the number of bits required for error correction. Without resolving this variation, the number of bits can still be substantially expanded by adding more insert lengths to take advantage of the modular system design.

From a computational perspective, we note that basecalling is getting faster and more reliable; however, raw signal classification is fundamentally a simpler problem than basecalling. Such classifiers can be trained with comparatively minimal data and expanded to nontraditional sensing; for example, our molbit approach could be easily extended to include nonstandard bases for additional security, but standard basecalling would not be possible without extensive amounts of training data. Similarly, future updates to nanopore sequencing chemistry would likely require retraining the classification model, made feasible by the low training data requirements. Even if a new chemistry modifies the raw signal to the extent that they are no longer classifiable via

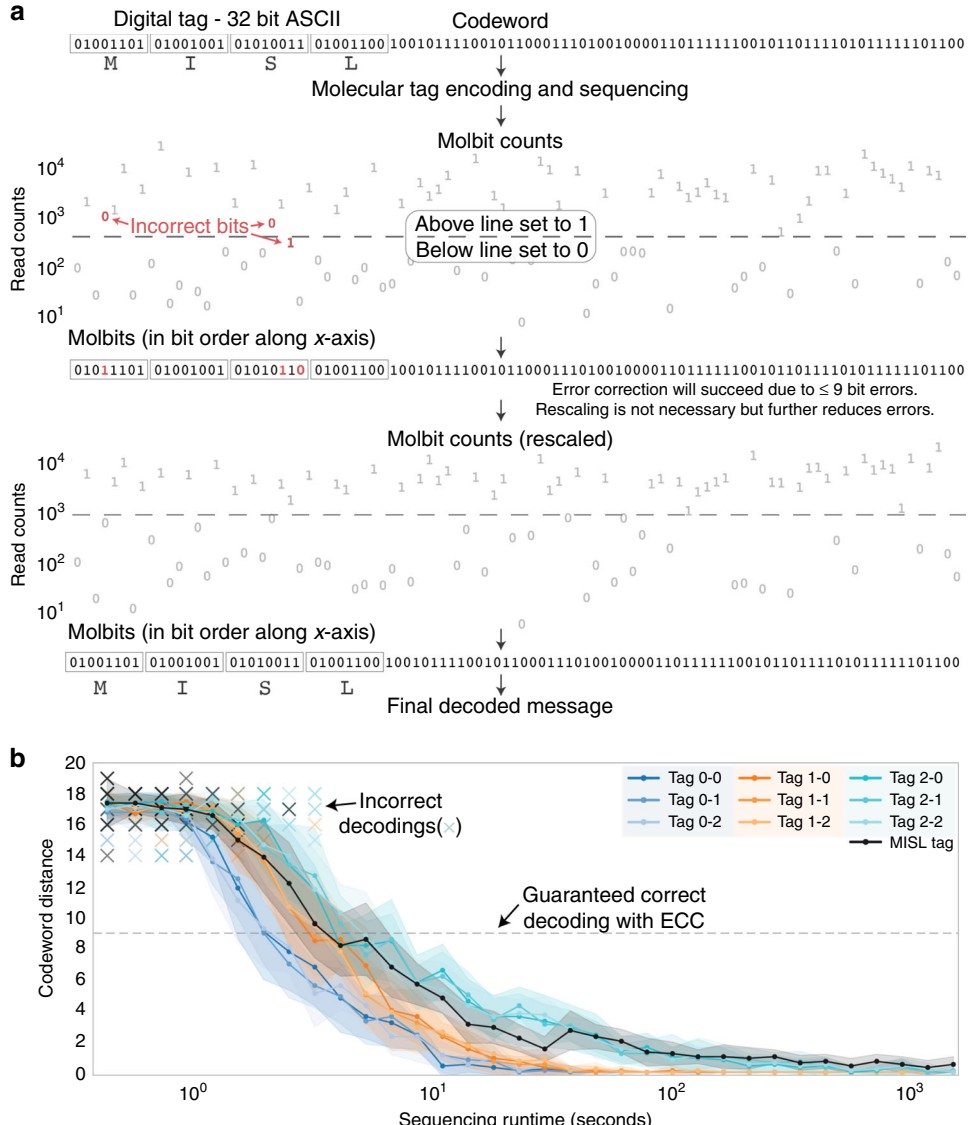

**Fig. 4 End-to-end data flow for the acronym "MISL". a** Encoding began by converting "MISL" to ASCII (32 bits). The digital tag was then multiplied by the 32 × 96-bit generator matrix, producing a 96-bit codeword. The codeword was converted to a molecular tag, stored, and sequenced for 35 min. When sequenced, molbits were identified and accumulated into a single count for each molbit. Bit errors are calculated with respect to the closest codeword, not necessarily the correct codeword. The dashed line represents the optimal threshold for binarizing these counts, which produced three incorrect bits. Rescaling the counts according to known read count variation reduced these errors, in this case eliminating errors entirely. **b** Minimum tag decoding distance as a function of sequencing runtime. Decoding occurred without a priori knowledge of the correct tag. Each color group (blue, orange, aqua, and black) represents a unique sequencing run, and each tag is a different remapping of a valid codeword to molbit labels. Bit errors are capped by the distance between tags, which is at least 18 in the chosen ECC. Incorrect decodings are marked with an X. At each time point, reads were sampled assuming an average of 10,000 reads per minute ($n = 10$); the S.D. of this sampling is shown as the shaded area. Source data are provided as a Source Data file.

raw signal, tags would still be readable by directly sequencing the DNA.

## Methods

**Molbit barcode design algorithm**. To produce highly separable squiggles, we used an evolutionary model. First, we initialized the 96 sequences. Fully random sequence initialization works fairly well; however, we chose to initialize using the output from a previous version of the molbit design algorithm, which produced slightly better results than random initialization after the molbits were further perturbed. This previous iteration was a brute force approach generating all repetitive sequences of length 2–6, based on the idea that a CNN may be able to use the periodicity to separate barcodes. However, this approach was limited for a few reasons: it did not account for sequence similarity, causing issues for labeling training data; any benefits to discriminability potentially provided by the periodicity were outweighed by mediocre sequence-based labeling; and requiring sequences to be periodic was too restrictive and significantly limited the space of

possible molbit barcodes. A similar "warm start" could be achieved by initializing many random sequences and choosing a subset of these that are maximally distinct within this random set, before beginning evolution. Starting sequences and their corresponding simulated nanopore squiggles are shown in Supplementary Fig. 4.

During mutation, we placed constraints on the sequences to ensure that they can be easily synthesized, assembled, and measured using the ONT MinION. If a mutated sequence does not fulfill these constraints, the mutation is reversed and attempted again. There are two types of constraints: (1) those that affect only a single sequence (independent constraints), and (2) those that impact the relationship between one sequence and all others (dependent constraints). Independent constraints require each sequence to be within a range of allowed GC content (30–70% GC), have a maximum folding potential (−8 kcal/mol) as calculated using NUPACK's MFE utility[13], exclude the BsaI cut site sequence (GGTCTC), and have a maximum homopolymer length of five for A/T and four for C/G. Dependent constraints require a minimum sequence dissimilarity, calculated using a local variant of the Smith–Waterman (SW) algorithm[12] (≤15 SW score; cost function +1 match, −1 mismatch, −8 gap); and a minimum squiggle

dissimilarity, calculated by simulating the sequence's nanopore squiggle using the Scrappie squiggler and computing the dynamic time warping similarity[14] for all squiggles vs. the new squiggle.

At the start of each round of sequence evolution, sequence order is randomized. Each sequence is mutated sequentially in this random order. The mutation is introduced by simultaneously modifying two adjacent nucleotides, in a random location. If the new sequence fails to fulfill the preceding constraints, we undo the mutation and try again until a maximum number of tries (100, arbitrarily), at which point we proceed to the next sequence. Next, we recalculate sequence similarities with respect to the new candidate sequence. If any sequences are too close to the new sequence, we undo the mutation and try again. Next, the nanopore squiggle is simulated for the new sequence using Scrappie squiggler. We recalculate the dynamic time warping similarity for all squiggles vs. the new squiggle, and, if any squiggles are now too close, we undo the mutation and try again. If the new mutation improves both the minimum and the average DTW dissimilarity between all squiggles, it is accepted; if not, the mutation is reversed and reattempted. Evolution ends when the optimization begins bouncing between just two sequences. At this point, the process has produced a local minimum as the result of a series of random incremental improvements, so further improvements may be gained only by significantly perturbing these final sequences. We show final sequences and their corresponding simulated nanopore squiggles in Supplementary Fig. 5.

**Molbit classification model**. We identify individual molbits using a classification model, which takes raw nanopore signals as input and outputs the molbit ID with an associated confidence. The model consists of a 5-layer CNN, followed by two fully connected layers with 50% dropout, and a final fully connected layer with softmax as the output layer. Each of the five CNN layers is identically structured, including a 1D convolutional layer with Relu activation, average pooling, and then batch normalization. We show a diagram of the model with exact parameters for each layer (e.g., kernel size) in Supplementary Fig. 1.

Ideally, we we would build a training data set by sequencing each of the 96 molbits separately. However, due to cost, we instead divided the 96 molbits into six runs of 16 molbits each. We constructed these sets to have a high predicted distance between the molbits within a set, meaning the most similar and easily confused molbits were not sequenced together for training data acquisition.

We assigned training labels using basecalling (Guppy version 3.2.2 with GPU acceleration) followed by Smith–Waterman sequence alignment (cost function: +1 match, −1 mismatch, −8 gap) against the full set of 96 molbits. We considered any SW score ≥15 to be a well-aligned match. As a quality measure, we also examined how many of these reads were labeled with one of the 16 possible molbits. An average of 98.7% ± 2.1% of well-aligned reads belonged to the true set of molbits across all training runs, indicating high quality labels.

After labeling the training data, we balanced the data set by allowing a maximum of 6000 reads occurrences for each molbit, with a total of 274,667 reads used for training. To pre-process the raw signal, we rescaled the signal using a median absolute deviation method modified from Oxford Nanopore Technologies' Mako classification tool, trimmed the signal to remove the variable-length stalled signal characteristic to the beginning of sequencing reads, and finally truncated the signal to the first 3000 data points. On average, the molbit barcode comprises ~10 to 15% of these 3000 observations. Due to stochastic variation in the dwell time of each nucleotide in the pore, there is some stretch in the signal, especially near the beginning of the read. This can cause considerable variation in both the position and length of the molbit barcode. This highlights the flexibility of the CNN, which allows us to be liberal with trimming since finding the exact end point of the barcode is not required.

We split the training data 85/15% to produce training and validation sets and trained the model for 109 iterations, with a final maximum training accuracy of 99.94% and validation accuracy of 97.78%. Confusion matrices for training and validating the final model are shown in Supplementary Fig. 6a.

We acquired and labeled testing data in the same manner as the training data, using two new sequencing runs, each containing a unique half of the molbits. Performance on these test sets was 98.1 and 95.7% for labeled data (Supplementary Fig. 6b). We washed and reused the flowcell from test set 1 for test set 2, which potentially contributed to a small portion of the errors present due to DNA carryover between the runs. We show read counts for these two test runs in Supplementary Fig. 7, noting when a molbit was possibly present from the previous run.

**Experimental methods**. We purchased forward and reverse strands of the 40 nt unique barcode sequences from Integrated DNA Technologies (IDT). Reverse strands contained a 5′ GAGT overhang and a 3′ dA-tail. We annealed forward and reverse strands by mixing them equimolar in 0.5 M PBS, boiling them at 94 °C for 2 min and then allowing them to cool at room temperature. To generate the insert spacer, we amplified an arbitrary 400 nt portion of plasmid pCDB180 [https://www.addgene.org/80677/] by PCR, using primers we designed to add BsaI cut sites to the ends of the amplified product. The primer sequences are available in Supplementary Table 1.

To assemble the molbits, 600 ng of the desired annealed barcodes (at equimolar concentrations) and 600 ng of the spacer were ligated together using NEB's Golden Gate Assembly Kit. We prepared molbits for sequencing using ONT's Ligation Sequencing Kit (SQK-LSK109) following the kit protocol (we skipped the "DNA repair and end-prep" step because molbits were already dA-tailed) and ONT's Flow

Cell Priming Kit (EXP-FLP001). Molbits were sequenced on a R9.4.1 MinION flow cell with bulk FAST5 raw data collection enabled on MinKNOW.

We dehydrated tags after nanopore adapter ligation by mixing the molbits with 1% trehalose dihydrate solution and lyophilizing the sample. To sequence, we rehydrated the lyophilized sample in nuclease-free water and carried out the sequencing run as described above.

**Reporting summary**. Further information on research design is available in the Nature Research Reporting Summary linked to this article.

## Data availability
The raw nanopore sequencing data sets generated and analyzed during the current study are available in the Harvard Dataverse repository with the identifier [https://doi.org/10.7910/DVN/RCLFNB][15]. Any other relevant data are available from the authors upon reasonable request. Source data are provided with this paper.

## Code availability
Code for analysis is available at https://github.com/uwmisl/Porcupine.

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

## Acknowledgements
We thank additional members of the Molecular Information Systems Lab for feedback on this work and also express our gratitude to Sergey Yekhanin (Microsoft) for helpful discussions about error correction codes. This project was funded in part by DARPA under the Molecular Informatics Program and gifts from Microsoft.

## Author contributions
K.D. participated in conceiving the idea and design/analysis of experiments, data curation, formal analysis, methodology, software, validation, visualization, writing—original draft, and writing—review and editing. K.Z. participated in investigation, validation, and writing—review and editing. A.M. contributed software for error analysis, as well as writing—review and editing. M.Q. contributed the ECC software and visualization, as well as writing—review and editing. K.S. participated in investigating the ECC strategy, in the design/analysis of experiments and supervised the work. L.C. participated in

conceiving the idea and design/analysis of experiments and supervised the work. J.N. participated in conceiving the idea and design/analysis of experiments, supervised the work, and contributed to writing—original draft and writing—review and editing.

## Competing interests

L.C., K.D., and J.N. declare that a provisional patent has been filed by the University of Washington, 16/879,214, "MOLECULAR TAGGING SYSTEM WITH NANOPORE-ORTHOGONAL DNA BARCODES", on May 20, 2020, covering aspects of this work including barcode design and readout. J.N. is a consultant to Oxford Nanopore Technologies. K.S. is a Microsoft employee. All other authors declare no competing interests.
