## [Peer Review File · Nature Communications]

Reviewers' Comments:

Reviewer #1:

Remarks to the Author:

Doroschak and colleagues present a method for creating DNA oligomer barcodes which are designed to be decoded using a nanopore DNA sequencer, except that the barcode information is read from the aggregate nanopore sensor output, rather than from the decoded DNA sequence. The identity of the tag is necessarily also contained in the sequence, but it would be valuable to design DNA barcodes which can be identified more quickly or cheaply when they do not necessarily require the full sequence. This is certainly an interesting, valuable, and relevant contribution to the field. I have some concerns which I hope can be addressed or clarified before it would be suitable for publication:

1. Can you quantify the benefits of using this type of barcode classifier, rather than simply using the sequence? It seems like you are implicitly using sequencing training data and tools in the MinION DNA sequencing ecosystem, so it is not immediately obvious why you wouldn't want to use the sequence data? It seems like looking for the "large" squiggle features would also have disadvantages in that you are not taking advantage of the much larger training datasets used to create the nanopore basecaller. I imagine there could be advantages in run time or coverage requirements, but these comparisons are not quantified in detail.
2. Calling the sequences "nanopore-orthogonal" seems like an overstatement?
3. The claim "without requiring a lab" seems difficult to define. Please be more specific.
4. The introduction claims that read/write cost is "not an issue" in archival storage applications. That seems like a bit of an overstatement?
5. The references and context in the field are lacking. Quite a few highly relevant and recent publications in this field are notably absent. In particular:

This meaning of the word "molbits" was introduced in:

- Cafferty, B.J., Ten, A.S., Fink, M.J., Morey, S., Preston, D.J., Mrksich, M. and Whitesides, G.M., 2019. Storage of information using small organic molecules. *ACS central science*, 5(5), pp.911-916.

Using random molecular codewords for error correction was recently described in:

- Arcadia, C.E., Kennedy, E., Geiser, J., Dombroski, A., Oakley, K., Chen, S.L., Sprague, L., Ozmen, M., Sello, J., Weber, P.M. and Reda, S., 2020. Multicomponent molecular memory. *Nature communications*, 11(1), pp.1-8.

There are also other recent works focusing on DNA barcoding of objects, on using raw squiggle barcode motifs from nanopore sequencers, and on encoding information in molecular form without using polymer sequences.

Below are two examples, but there are surely more. Please expand the references.

- Han, R., Wang, S. and Gao, X., 2020. Novel algorithms for efficient subsequence searching and mapping in nanopore raw signals towards targeted sequencing. *Bioinformatics*, 36(5), pp.1333-1343.

- Koch, J., Gantenbein, S., Masania, K., Stark, W.J., Erlich, Y. and Grass, R.N., 2020. A DNA-of-things storage architecture to create materials with embedded memory. *Nature Biotechnology*, 38(1), pp.39-43.

6. The caption of Figure 1 says that "Software then identifies the tag, either with or without knowledge of the original tag." What does this mean?
7. In Fig 2b, you show that you can design a sequence to match a target squiggle shape. But later you seem to give up this approach, and instead randomly mutate the sequence. Following from the Fig 2b capability, why wouldn't you first generate a library of maximally-separable squiggles, and then create the DNA sequences which produce these squiggles? Can you clarify the necessity of the random evolution?
8. What does it mean that you "chose to initialize using a previous iteration of molbit barcodes"? In Supplementary Figure 4, many of these starting sequences appear to be quite periodic. Where did this previous iteration come from?
9. Please briefly describe "Golden Gate" assembly, and provide a citation when it is introduced.
10. Why did you choose not to explore the spacer length encoding?
11. Reporting "a 42.6% reduction" in error rate seems overly precise, especially given that the confidence intervals of the two examples overlap.
12. Is there any risk introduced from doing the sequencing prep far ahead of time, before labeling the object and recovering the sample? Are the tags potentially less efficient during the nanopore capture process, or potentially corrupted in different ways? Could this potentially have any effect on the low read counts?
13. You write that you "decoded the tag by binarizing the counts with varying thresholds". Can you clarify which thresholds this refers to, and how they were chosen? How do you infer the "most likely correct decoding"?
14. Why is it valuable to have "visually unique" traces?
15. How will this method be affected by future updates to nanopore sequencing chemistry?

Reviewer #2:

Remarks to the Author:

In the manuscript, "Porcupine: Rapid and robust tagging of physical objects using nanopore-orthogonal DNA strands", Doroschak et al. describe their generation and subsequent classification of DNA-based tags encoded as "molbits" which are easily distinguishable not only in sequence space, but also in current space. This is a great strategy and a powerful tool not only for the authors stated interests of DNA barcoding of physical objects, but also for DNA data storage and even traditional sample barcoding. Generally, I like the work, but I had some points that would improve it and some points that need clarification.

1. Saying that you can do this work (without a lab) is a little misleading - perhaps instead saying that you could do it in a low resource/minimal lab environment. Even if you just need a minION and have dehydrated the libraries.
2. First paragraph: The method being presented is a strategy for labeling physical objects yet this paragraph mostly discusses methods of encoding large data in DNA. This is evident through the paper, where the authors go a bit back and forth on what the primary use of the work is - data storage, or just tagging (a form of storage, sure, but lower info density). I would add I think it's a missed opportunity to more clearly integrate/expand a discussion of the possibility of molbits for more accurate or better multiplexing, especially given the higher yield of recent nanopore sequencing work. Multiplexing efficiencies in nanopore are still quite poor (cf Deepbinner,

porechop, etc.) and more tools to improve it with "nanopore-orthogonal tags" could be quite valuable.

3. The intro closes with claims that porcupine can be used for sample barcoding-- this claim would be really strengthened by demonstrating this use. I know that this is unlikely in the current environment, but just simply demonstrating it or perhaps a simple comparison of how this stands against current demultiplexing efficiencies would be powerful.

4. The description of how the molbits are mixed/identified is not clear - the figure is a little better - but the fact that you are encoding a 96-bit value by the presence or absence of the molbits was not immediately obvious. Suggest some more clarity in the text.

5. What the spacer is made of should be specified in the main text - I had to hunt for it.

6. It seems that length is not the most robust separation metric by itself - did the authors look into the sequence to determine what happened to the 1600bp? Presumably this was DNA fragmentation in the library prep - basecalling and analysis of the signal to clarify this is warranted.

7. Length based coding especially seems like it would be an issue for the planned dehydration - presumably that would lead to even further fragmentation?

8. For classifying the molbits, Supp Fig1 indicates the first 3000 data points in the trimmed signal time series were used, what fraction of this signal is the molbit versus the spacer?

9. It seems based on the correction the authors have to do for the nonzero read counts and variation in counts for the different molbits that there is poor performance from some of the molbits? Is it possible to just pick the best and go down to a 64-bit encoding? A computational analysis would answer this. Would that actually improve the result? In any case, further clarification of this point would aid in clarity.

10. Further - is the variation due to the strand-strand hybridization and overhang? Is the overhang (GATG) present or less present in the molecules with more severe variation? Further - how are the authors quantifying their sample before combining the bits? I agree with the authors that the variation is not critical if reproducible, but it is puzzling.

11. Saying that you "simulated" the runs (Fig 4b) seems to be a bit confusing - I believe in this case you were saying that in already existing datasets, you could look at the time of sequencing and determine when you had enough data to decode the correct tag? This could be phrased more clearly. Are you simulating just the sequencing run - or also simulating the software at the same time? It's also important to note that the library preparation takes a non zero amount of time.

12. The dehydration in Supplementary Figure 3 is very impressive - I'm especially surprised the nanopore motor protein survived in the dehydrated form for 4 weeks? But again here I note what appears to be

Point-by-point response to reviewers

“Porcupine: Rapid and robust assembly and decoding of molecular tags with DNA-based nanopore signatures”

We thank the reviewers for their numerous constructive comments and suggestions. Below we provide a point-by-point response to specific comments and highlight changes that have been made to the manuscript. Each change in the text is also labeled in the margin with the specific point that was addressed.

Referee #1

1. Can you quantify the benefits of using this type of barcode classifier, rather than simply using the sequence? It seems like you are implicitly using sequencing training data and tools in the MinION DNA sequencing ecosystem, so it is not immediately obvious why you wouldn't want to use the sequence data? It seems like looking for the “large” squiggle features would also have disadvantages in that you are not taking advantage of the much larger training datasets used to create the nanopore basecaller. I imagine there could be advantages in run time or coverage requirements, but these comparisons are not quantified in detail.

We found that our signal-based classification method was able to confidently identify ~30% more reads than our sequence-based method. This subsequently presented challenges for validation and data labeling (i.e., how can we confirm the correctness of these labels if we do not have quality basecalls/alignments as a ground truth?). We attempted to reason about their correctness by correlating the read counts from the additional 30% of reads with the sequence-labeled reads, and found a strong correlation, indicating that these are likely not spurious calls (Supplementary Figure 2). In the text, this was previously worded in a way that did not call attention to this specific advantage, so we have now done so.

To address the “signal vs. sequence” question more broadly, we note that classification is an inherently simpler problem than full sequence decoding / basecalling, where the problem can be reduced to distinguishing between 96 distinct signals rather than reproducing the exact series of underlying nucleotides generating an arbitrary nanopore signal. We have added this brief explanation to the section introducing the CNN (see pages 7-8).

2. Calling the sequences “nanopore-orthogonal” seems like an overstatement?

We agree that calling the sequences “nanopore-orthogonal” could be misleading. We have modified this in the title as well as all occurrences in the abstract, introduction, and methods sections.

The new title is “Porcupine: Rapid and robust assembly and decoding of molecular tags with DNA-based nanopore signatures.”

3. The claim “without requiring a lab” seems difficult to define. Please be more specific.

We agree with both reviewers here that we should have been more clear about the resources required to use this system, and also avoiding overstating the “lab-free” claim. We changed the text to read “low resource environment,” and also added a sentence describing the requirements for writing and reading tags at the end of the introduction. (Abstract and page 5).

4. The introduction claims that read/write cost is “not an issue” in archival storage applications. That seems like a bit of an overstatement?

We have reworked the introduction to focus more on molecular tagging and raw signal manipulation, incorporating the references in #5. Our initial introduction was an attempt to contrast large-scale data storage with low-density information in molecular tagging.

5. The references and context in the field are lacking. Quite a few highly relevant and recent publications in this field are notably absent. In particular:

This meaning of the word “molbits” was introduced in:

- Cafferty, B.J., Ten, A.S., Fink, M.J., Morey, S., Preston, D.J., Mrksich, M. and Whitesides, G.M., 2019. Storage of information using small organic molecules. ACS central science, 5(5), pp.911-916.

Using random molecular codewords for error correction was recently described in:

- Arcadia, C.E., Kennedy, E., Geiser, J., Dombroski, A., Oakley, K., Chen, S.L., Sprague, L., Ozmen, M., Sello, J., Weber, P.M. and Reda, S., 2020. Multicomponent molecular memory. Nature communications, 11(1), pp.1-8.

There are also other recent works focusing on DNA barcoding of objects, on using raw squiggle barcode motifs from nanopore sequencers, and on encoding information in molecular form without using polymer sequences.

Below are two examples, but there are surely more. Please expand the references.

- Han, R., Wang, S. and Gao, X., 2020. Novel algorithms for efficient subsequence searching and mapping in nanopore raw signals towards targeted sequencing. Bioinformatics, 36(5), pp.1333-1343.

- Koch, J., Gantenbein, S., Masania, K., Stark, W.J., Erlich, Y. and Grass, R.N., 2020. A DNA-of-things storage architecture to create materials with embedded memory. Nature Biotechnology, 38(1), pp.39-43.

In the process of reworking the introduction, we have added more context that references these papers, plus several others in the areas of DNA tagging and raw nanopore signal processing.

6. The caption of Figure 1 says that “Software then identifies the tag, either with or without knowledge of the original tag.” What does this mean?

This means that no a priori knowledge of the digital tag is required to decode a molecular tag. We have clarified the figure caption accordingly.

7. In Fig 2b, you show that you can design a sequence to match a target squiggle shape. But later you seem to give up this approach, and instead randomly mutate the sequence.

We intended Fig. 2b to demonstrate that scrappie is a useful tool for simulating squiggles, rather than displaying a method to algorithmically design a sequence to match a target squiggle shape. The sequence producing the nanopore squiggle “UW” was hand designed as a simple demonstration. We have updated the text to clarify that it was hand-designed (page 5).

Following from the Fig 2b capability, why wouldn’t you first generate a library of maximally-separable squiggles, and then create the DNA sequences which produce these squiggles?

Generating a library of maximally-separable squiggles and then identifying possible underlying DNA sequences may not be feasible due to physical limitations of the signals produced by DNA sequences. First, even when constrained by a reasonable signal dynamic range, not all drawn lines can be manifested as squiggles produced by DNA in the real world. The fixed number of transitions between kmers (e.g. $ACTAG \rightarrow CTAG\{A|C|G|T\}$) severely restricts valid combinations of simulated data points. Furthermore, other required sequence characteristics like folding potential and sequence similarity must still be controlled for, and a signal-first approach shifts the burden of checking these characteristics to after signal design, meaning even squiggles with valid generating DNA sequences may not be viable molbits.

Can you clarify the necessity of the random evolution?

Random evolution is not inherently necessary, but is just one method to gradually force sequences to be more distinguishable. A hybrid approach, which we express as a direction for future work, could include a generative model in which sequences would be designed concurrently rather than perturbed individually, but sequence characteristics could be accounted for as a constraint in the model.

8. What does it mean that you “chose to initialize using a previous iteration of molbit barcodes”? In Supplementary Figure 4, many of these starting sequences appear to be quite periodic. Where did this previous iteration come from?

The previous iteration came from a brute force approach finding the most separable set of sequences out of all repetitive sequences of repeat length 2-6 (excluding some due to GC content, homopolymers, BsaI

cut site, and folding potential). The idea was that a CNN may be able to use the periodicity to separate barcodes. Instead, we found that straying from this approach proved to be better.

This approach was limited for a few reasons: it did not account for sequence similarity, causing issues for labeling training data; any benefits to discriminability potentially provided by the periodicity were outweighed by mediocre sequence-based labeling; and requiring sequences to be periodic was too restrictive and significantly limited the space of possible molbit barcodes. However, since these sequences were quite separable on the signal level, we chose to give the evolutionary algorithm a “warm start” by using sequences generated by the previous version.

We have incorporated the above explanation and a suggestion for another form of “warm start” initialization in the first paragraph of Materials and Methods (page 15).

9. Please briefly describe “Golden Gate” assembly, and provide a citation when it is introduced.

This change has been made on page 5.

10. Why did you choose not to explore the spacer length encoding?

We chose this approach to reduce experimental complexity. We had explored length to a greater extent with previous molbit sets and felt that it was straightforward, so here we demonstrated the concept and moved on to the more conceptually interesting component. We added a brief note of explanation at the end of the paragraph for describing providing context about this choice (see page 7).

11. Reporting “a 42.6% reduction” in error rate seems overly precise, especially given that the confidence intervals of the two examples overlap.

We agree and have edited the text to use a more appropriate number of significant figures in several places on pages 7, 8, and 9.

12. Is there any risk introduced from doing the sequencing prep far ahead of time, before labeling the object and recovering the sample?

Based on our results examining the quality, length, and yield of the reads, we believe that there is minimal risk to carrying out sequencing prep far ahead of time, at least as long as 4 weeks ahead (Supplementary Figure 3). We envisioned that sequencing preparation would be carried out immediately before dehydration to maximize tag stability. We added this as a point of discussion on page 11.

Are the tags potentially less efficient during the nanopore capture process, or potentially corrupted in different ways?

Tag reading efficiency can be measured by the sequencing yield over time, which was stable across all three measured conditions (fresh, and dehydrated at 0 and 4 weeks). We are unsure about other forms of corruption that would not be adequately measured by the yield, Q-scores, or sequence length.

Could this potentially have any effect on the low read counts?

This advance preparation is unlikely to be the cause of the low read counts, as most runs were conducted immediately after sample preparation.

13. You write that you “decoded the tag by binarizing the counts with varying thresholds”. Can you clarify which thresholds this refers to, and how they were chosen? How do you infer the “most likely correct decoding”?

This refers to the read count threshold by which presence or absence is determined. We have clarified this in the text (see page 9) and the legend for Figure 3.

To contextualize this response, we added that correct decoding is guaranteed by this particular ECC at or below 9 bit errors (previous paragraph; also page 9).

14. Why is it valuable to have “visually unique” traces?

Since nanopore signals are one-dimensional, signals that are visually distinct are also more easily separable by direct signal comparison and classification methods, like DTW and CNNs.

15. How will this method be affected by future updates to nanopore sequencing chemistry?

Updates to nanopore sequencing chemistry would likely negatively impact the raw signal-based classification of existing molbits; however, due to the relatively low training data requirements, retraining the model would be feasible. Presumably any changes to the sequencing chemistry would not make nucleotides more difficult to distinguish, but if necessary, new molbits could be generated using the same molbit design approach, provided that there is still a way to simulate hypothetical nanopore raw current. (See page 11.)

Referee #2

1. Saying that you can do this work (without a lab) is a little misleading - perhaps instead saying that you could do it in a low resource/minimal lab environment. Even if you just need a minION and have dehydrated the libraries.

We agree with both reviewers here that we should have been more clear about the resources required to use this system, and also avoiding overstating the “lab-free” claim. In two instances (Abstract and p. 5), we changed the text to read “low resource environment,” and also added a sentence describing the requirements for writing and reading tags in the introduction. We thank the referee for their suggestion of wording.

2. First paragraph: The method being presented is a strategy for labeling physical objects yet this paragraph mostly discusses methods of encoding large data in DNA. This is evident through the paper, where the authors go a bit back and forth on what the primary use of the work is - data storage, or just tagging (a form of storage, sure, but lower info density). I would add I think it's a missed opportunity to more clearly integrate/expand a discussion of the possibility of molbits for more accurate or better multiplexing, especially given the higher yield of recent nanopore sequencing work. Multiplexing efficiencies in nanopore are still quite poor (cf Deepbinner, porechop, etc.) and more tools to improve it with “nanopore-orthogonal tags” could be quite valuable.

We have reworked the introduction to focus more on molecular tagging and raw signal manipulation, including multiplexing. (See also our response to #3 below.)

3. The intro closes with claims that porcupine can be used for sample barcoding-- this claim would be really strengthened by demonstrating this use. I know that this is unlikely in the current environment, but just simply demonstrating it or perhaps a simple comparison of how this stands against current demultiplexing efficiencies would be powerful.

Unfortunately, this is not something we were able to accomplish due to ongoing lab closures. Instead, in the introduction when discussing demultiplexing tools, we emphasize that these tools could likely be improved by designing barcodes specifically for the task at hand by making the raw signals more differentiable. Because we have not tested this directly, especially in the context of larger sample diversity, we feel we cannot strengthen this claim beyond suggesting it as a potential use case or using raw signal demultiplexing papers as broader context for raw nanopore signal processing.

4. The description of how the molbits are mixed/identified is not clear - the figure is a little better - but the fast that you are encoding a 96-bit value by the presence or absence of the molbits was not immediately obvious. Suggest some more clarity in the text.

We have reworded the section of the introduction that first describes Porcupine (see page 3).

5. What the spacer is made of should be specified in the main text - I had to hunt for it.

We have added this in the main text on page 7.

6. It seems that length is not the most robust separation metric by itself - did the authors look into the sequence to determine what happened to the 1600bp? Presumably this was DNA fragmentation in the library prep - basecalling and analysis of the signal to clarify this is warranted.

The majority of the short strands in the 1600 bp sample can be explained by fragmentation, supported by an evaluation of basecalled, aligned sequences. After aligning all basecalled sequences to the insert fragment sequence using BWA-MEM [1], we examined reads that were shorter than the signal length threshold (i.e., those mislabeled as 400 bp). We found that 28% of these reads were truncated, meaning they aligned well to the beginning of the insert but terminated prematurely at various lengths, indicating fragmentation or sequencing failures. Another 59% mapped the majority of the read to a random portion of the reference, indicating fragmentation. The remaining reads were simply poor quality reads that did not basecall and/or map well. To mitigate this, a length purification step could be added, making preparation longer to provide a higher quality readout. We made a note of this on page 7 and discussed further in the Supplementary Text (page 19).

7. Length based coding especially seems like it would be an issue for the planned dehydration - presumably that would lead to even further fragmentation?

DNA fragmentation could be an issue for length-based encoding over long periods of time. Anchordoquy et. al examined preservation of plasmid DNA in a variety of circumstances, including lyophilization in trehalose at room temperature over a period of 24 weeks [2]. This study demonstrated that while lyophilized DNA is initially stable, single strand breakages occurred in ~40% of plasmids over a period of 24 weeks at room temperature. Despite this potential for breakage, we feel that especially if the time of dehydration is known, length-based encoding errors could potentially be remedied via error correction, taking advantage of the one-sided nature of the errors (since long strands can only become shorter, and not vice versa).

8. For classifying the molbits, Supp Fig1 indicates the first 3000 data points in the trimmed signal time series were used, what fraction of this signal is the molbit versus the spacer?

The molbit barcode comprises roughly 15% of the total raw signal window (length 3000) input to the CNN. The position of the molbit varies due to stochastic variation in the dwell time for each kmer/nucleotide in the pore, particularly at the beginning of the signal when the sequencing adapter is beginning to unwind. This highlights the flexibility of the CNN, which allows us to be liberal with

trimming since the classifier does not need to be sensitive to the specific start and endpoints of the molbit barcode. We have added this information to the supplement (page 16).

9. It seems based on the correction the authors have to do for the nonzero read counts and variation in counts for the different molbits that there is poor performance from some of the molbits? Is it possible to just pick the best and go down to a 64-bit encoding? A computational analysis would answer this. Would that actually improve the result? In any case, further clarification of this point would aid in clarity.

We looked into the possibility of reducing our collection to just the most reliable molbits. Although ideally we would like to see molbits that have uniform read counts when set to 1, as long as a molbit's counts can be corrected, it is not necessarily significantly less performant than the others.

As the number of total molbits drops, each must become more reliable in order to maintain the desired chance of incorrect decoding (new Supplementary Figure 10). Another way to think about this tradeoff is to consider that even unreliable molbits are still conveying useful information that the ECC is able to use to deduce the corrected message. For example, a collection of 96 molbits can reliably encode a 32-bit message even if each molbit is wrong 3% of the time on average. But if we reduce our pool slightly to just 90 molbits, then they can only be wrong 2% of the time on average. At 64 molbits, a dramatic reduction in the corruption rate would be required, down to the order of ~0.15%.

We have added this discussion to the Supplementary Text section on Error correction (page 20).

10. Further - is the variation due to the strand-strand hybridization and overhang? Is the overhang (GATG) present or less present in the molecules with more severe variation?

We designed the molbit barcode sequences to avoid strand-strand hybridization as much as possible, but it could still feasibly be occurring to some extent.

While examining the sequence contents in response to #6, we discovered an error in the overhang sequence used for ligation of the insert to molbits. We intended to use the overhang pair ACTC/GAGT, and ordered our molbit sequences with the overhang GAGT. However, we mistakenly used insert sequences from an older version, with the overhang GCTG instead of ACTC. Surprisingly, the Golden Gate assembly worked with enough efficiency that we did not notice, and we had previously only examined either the barcode or the insert sequence contents separately. We have checked for occurrences of both ACTC and GCTG in the molbit sequences and found no significant correlation between occurrence of these motifs and molbit read count variation.

We added plots to Supplementary Figure 8a comparing the number of occurrences of GCTG and ACTC in each molbit barcode to its read counts. The overhang sequences were not more or less present in strands with the most variation.

We have also updated the Methods and Materials section to reflect the true overhangs used.

Further - how are the authors quantifying their sample before combining the bits? I agree with the authors that the variation is not critical if reproducible, but it is puzzling.

We did not quantify before combining bits, but rather mixed molbit barcodes in equimolar amounts and then ligated them to the insert all at once. In retrospect, we could have separately assembled and quantified some of the worst offending bits (e.g. those in Supplementary Figure 8b), but due to lab closures we are unable to complete this analysis.

11. Saying that you “simulated” the runs (Fig 4b) seems to be a bit confusing - I believe in this case you were saying that in already existing datasets, you could look at the time of sequencing and determine when you had enough data to decode the correct tag? This could be phrased more clearly. Are you simulating just the sequencing run - or also simulating the software at the same time?

When simulating runs, we are not only examining the run time, but also reassigning molbit labels so they encode a different codeword (e.g. molbit 0 in the original tag is given a new label of molbit 42 in the simulated tag). Since the codewords are not all the same distance from one another (the minimum edit distance between tags is 18, but many codewords have a minimum edit distance to the nearest valid codeword of 20+), this can impact the error rate both positively and negatively when decoding.

Additionally, after “simulating” these tags by remapping the molbit labels, we simulated the sequencing run by subsampling the reads to approximate different runtimes. This allows us to estimate the amount of data required to decode the correct tag.

We have reworked the corresponding section in the text to clarify; see page 9.

It’s also important to note that the library preparation takes a non zero amount of time.

We added a brief note on page 11, emphasizing that readout time is only reduced by front-loading the library preparation at the time of tag creation.

12. The dehydration in Supplementary Figure 3 is very impressive - I’m especially surprised the nanopore motor protein survived in the dehydrated form for 4 weeks? But again here I note what appears to be differences in read length - truncation occurring in the fresh sample? Could the authors elaborate on the size differences observed, especially if a Q-score filter is applied?

We concur with the referee’s surprise that the motor protein survived in a dehydrated form for this long. We note that the slight difference between the fresh and dehydrated tags is most likely due to experimental variation from tag mixing and/or library preparation; the fresh tag was prepared and run first, and then months later, the dehydrated tag library was prepared and split after dehydration. Since the

dehydrated runs show more similarity to each other than to the fresh sample, it is likely that the variation produced by degradation over the four week time span is smaller than the variation produced by sample preparation. We have added a note in the caption of Supplementary Figure 3 for context. To examine the size differences between the samples, we have provided histograms of the read lengths for different ranges of Q-scores (added to Supplementary Figure 3).

References

[1] Li, H. Aligning sequence reads, clone sequences and assembly contigs with BWA-MEM(2013). <http://arxiv.org/abs/1303.3997>.

[2] Anchordoquy, T. J. & Molina, M. C. Preservation of DNA. Cell Preservation Technology **5**, 180–188 (2007).

Reviewers' Comments:

Reviewer #1:

Remarks to the Author:

I appreciate the authors' clear and helpful responses, and the corresponding updates to the manuscript. All of my previous comments have been addressed. This is a very nice work and it appears ready for publication.

Reviewer #2:

Remarks to the Author:

The authors have addressed my comments and the work is suitable for publication.